# Application of a Physiologically Based Pharmacokinetic Model to Develop a Veterinary Amorphous Enrofloxacin Solid Dispersion

**DOI:** 10.3390/pharmaceutics13050602

**Published:** 2021-04-22

**Authors:** Kaixiang Zhou, Meixia Huo, Wenjin Ma, Kun Mi, Xiangyue Xu, Samah Attia Algharib, Shuyu Xie, Lingli Huang

**Affiliations:** 1National Reference Laboratory of Veterinary Drug Residues (HZAU) and MAO Key Laboratory for Detection of Veterinary Drug Residues, Huazhong Agricultural University, Wuhan 430070, China; flyingkai@webmail.hzau.edu.cn (K.Z.); HuoMeixia@webmail.hzau.edu.cn (M.H.); mawenjin@webmail.hzau.edu.cn (W.M.); mikun@webmail.hzau.edu.cn (K.M.); xuxiangyue@webmail.hzau.edu.cn (X.X.); samah.alghareeb@fvtm.bu.edu.eg (S.A.A.); xieshuyu@mail.hzau.edu.cn (S.X.); 2Department of Clinical Pathology, Faculty of Veterinary Medicine, Benha University, Moshtohor, Toukh 13736, Egypt; 3MOA Laboratory for Risk Assessment of Quality and Safety of Livestock and Poultry Products, Huazhong Agricultural University, Wuhan 430070, China

**Keywords:** intestinal infections, enrofloxacin, solid dispersion, physiologically based pharmacokinetics model, formulation strategies, dose design

## Abstract

Zoonotic intestinal pathogens threaten human health and cause huge economic losses in farming. Enrofloxacin (ENR) shows high antibacterial activity against common intestinal bacteria. However, its poor palatability and low aqueous solubility limit the clinical application of ENR. To obtain an ENR oral preparation with good palatability and high solubility, a granule containing an amorphous ENR solid dispersion (ENR-SD) was prepared. Meanwhile, a PBPK model of ENR in pigs was built based on the physiological parameters of pigs and the chemical-specific parameters of ENR to simulate the pharmacokinetics (PK) of ENR-SD granules in the intestinal contents. According to the results of parameter sensitivity analysis (PSA) and the predicted PK parameters at different doses of the model, formulation strategies and potential dose regimens against common intestinal infections were provided. The DSC and XRD results showed that no specific interactions existed between the excipients and ENR during the compatibility tests, and ENR presented as an amorphous form in ENR-SD. Based on the similar PK performance of ENR-SD granules and the commercial ENR soluble powder suggesting continued enhancement of the solubility of ENR, a higher drug concentration in intestinal contents could not be obtained. Therefore, a 1:5 ratio of ENR and stearic acid possessing a saturated aqueous solubility of 1190 ± 7.71 µg/mL was selected. The predictive AUC_24h_/MIC_90_ ratios against *Campylobacter jejuni*, *Salmonella*, and *Escherichia coli* were 133, 266 and 8520 (>100), respectively, suggesting that satisfactory efficacy against common intestinal infections would be achieved at a dose of 10 mg/kg b.w. once daily. The PSA results indicated that the intestinal absorption rate constant (Ka) was negatively correlated with the C_max_ of ENR in the intestine, suggesting that we could obtain higher intestinal C_max_ using P-gp inducers to reduce Ka, thus obtaining a higher C_max_. Our studies suggested that the PBPK model is an excellent tool for formulation and dose design.

## 1. Introduction

Intestinal bacterial infections caused by zoonotic pathogens, including *Campylobacter jejuni* (*Camp. jejuni*), *Salmonella* spp. and *Escherichia coli* (*E. coli*), have resulted in serious threats to human health and enormous economic losses in farming [1,2,3,4]. The economic losses owing to salmonellosis caused by *Salmonella* spp. in the US exceed an estimated $3.5 billion per year [5]. The infection rate of *Salmonella enterica* in finishing pigs is as high as 32.9% [6]. Moreover, these zoonotic intestinal bacteria can be transmitted to humans through food, water, and feces. Reportedly, pork is the most common food vehicle for *Salmonella Typhimurium* outbreaks [7]. It is vital to prevent and control the emergence of intestinal infections in livestock and poultry to reduce the threats of intestinal zoonotic bacteria to humans.

Enrofloxacin (ENR), an animal-specific antibiotic, is commonly used in farming because of its favorable pharmacokinetic (PK) profile, strong bactericidal activity and high post-antibacterial effect [8,9]. The 50% minimum inhibitory concentrations (MIC_50_) of ENR against *Camp. jejuni*, *Salmonella* spp. and *E. coli* were 8.0 µg/mL, 1.0 µg/mL, and 0.5 µg/mL, respectively [10,11,12]. However, due to the bitter taste of ENR [13,14], it can only be administered to pigs by the intramuscular route at present (Animal Drug @FDA). The oral route is undoubtedly the best route to treat intestinal infections, and it is also the most preferred route in the veterinary clinic [15,16]. Additionally, the saturated solubility of ENR in neutral conditions is poor (~490 µg/mL, 37 °C, pH = 7.4) [17], and recrystallization may occur when dissolved ENR is emptied from the stomach to the intestine, resulting in a low concentration of ENR in the intestinal contents. Thus, redissolution is related to delaying the peak time (T_max_) and reducing the peak concentration (C_max_) of ENR in intestinal contents. Therefore, developing a new oral ENR product for pigs with good palatability and high solubility to improve its prevention and control level against intestinal pathogens has become a valuable and urgent goal.

Some researchers have attempted to improve the palatability of ENR [9]. Our previous studies proved that combining enteric coating with solid lipid nanoparticles could help to obtain taste-masking granules for pigs. Compared with commercial soluble powder, the oral bioavailability of the ENR enteric granules was improved by 264% [18]. Due to their efficient oral absorption, enteric granules have been designed for lung infections (e.g., *Mycoplasma* pneumonia and pleuropneumonia). Although enteric coating technology is helpful for palatability improvement, because of the surface coating, the dissolution of enteric granules in the gastrointestinal tract is limited, thus leading to a relatively lower C_max_ and longer T_max_ values in the intestine. ENR exhibits concentration-dependent characteristics, and its antibacterial effect depends on the AUC_24h_/MIC [18]. Therefore, a technology that could improve both the palatability and solubility of ENR was needed in the present study.

In a solid dispersion (SD), the drug is molecularly amorphous and/or microcrystalline forms dispersed in a solid lipid [19,20]. Because of the high dissolution of amorphous drugs, amorphous drug SD has become one of the common technologies for the solubility and dissolution rate enhancement of poorly water-soluble drugs [21]. Guo et al. [22] demonstrated that compared with torcetrapib powder, torcetrapib-copovidone amorphous SD significantly improved the solubility and C_max_ of torcetrapib in the plasma by approximately 5- and 11-fold, respectively. In addition, the taste-masking effect and market future of amorphous drug SDs are also expected [23]. Therefore, this paper attempts to apply amorphous SD technology to improve the palatability and solubility of ENR, thus achieving its higher efficacy against infections caused by common intestinal bacteria.

PK studies are essential for formulation evaluation and dose design. However, a number of experimental animals are needed to clarify the PK characteristics and dose regimen design. The physiologically based pharmacokinetics (PBPK) model is a mechanistic model that can simulate the real-time dynamic processes of drugs in different organs at different doses by incorporating the physiological parameters of animals and the physicochemical parameters of drugs. PBPK models have great extrapolation and prediction power and thus have attracted increasing attention in the formulation design and clinical trials of new drugs. Reportedly, PBPK models have been applied to predict the bio-performance of amorphous drug SDs [24,25]. Meanwhile, PBPK models were used to design the dose regimen by combining the PK profile with the pharmacodynamics (PD) value [26,27,28]. For instance, Schuck et al. [29] adopted the AUC_24h_/MIC ratio to evaluate the efficacy of ciprofloxacin immediate-release and extended-release preparations. Therefore, the AUC_24h_/MIC method was adopted in this study to predict the potential dose of ENR-SD granules for common zoonotic intestinal pathogens.

The present study developed an amorphous ENR-SD with high aqueous saturated solubility and good palatability for pigs, and then a PBPK model for ENR in pigs was built. After the prediction accuracy of the PBPK model was verified, the PK profiles of ENR-SD granules in the intestinal contents versus time at different doses were predicted. Then, the potential doses focused on several common zoonotic intestinal pathogens were predicted by combining the PK data and the MIC_90_ value of ENR. Meanwhile, further formulation strategies were provided based on parameter sensitivity analysis (PSA). We emphasized that the PK curves of ENR-SD granules in the intestinal contents at different doses were predicted by the PBPK model directly rather than animal experiments; thus, potential dose regimens against common intestinal infections and further formulation strategies of ENR-SD granules were provided by the PBPK model without animal use. This is the first study to apply the PBPK model to the field of new veterinary drug development, which will be beneficial for simplifying the regulation and development processes of new veterinary drugs.

## 2. Materials and Methods

### 2.1. Materials

ENR standard (content: 99.0%) was provided by Dr. Ehrenstorfer Gmbh. Native ENR (content: ≥96.0%) was purchased from Jinxin Pharmaceutical (Zhejiang, China). Commercial ENR tablets (approved for pets, content: 25 mg/tablet) were bought from Nanhai Eastern Along Pharmaceutical CO., Ltd. (Foshan, China). Stearic acid was provided by CHINEWAY (Shanghai, China). Sodium carboxymethyl cellulose (CMCC-Na), NaCl, HCl, and NaOH were provided by Sinopharm Group Chemical Reagent Co., Ltd. (Shanghai, China). Pepsin (1:10,000) and corn starch were purchased from Aladdin (Shanghai, China).

### 2.2. Animal

Eighteen clinically healthy three-way hybrid pigs (25 ± 5 kg) were provided by Jinling pig farm (Wuhan, China). The pigs were fed at laboratory animal rooms at the National Reference Laboratory of Veterinary Drug Residues (HZAU). They were fed with drug-free feed and drinking for seven days. The environment was kept at a suitable relative humidity (45–65%) and temperature (18–25 °C), respectively. All the experimental protocols were approved by the Institutional Animal Care and Use Committee at Huazhong Agricultural University (Approval number: HZAUSW-2019-024, data: August 2019) and followed the guidelines of Hubei Science and Technology.

### 2.3. Preparation and Evaluation of ENR-SD Granules

#### 2.3.1. Compatibility Test

Thermal analysis and optics methods were adopted to study the compatibility between ENR and excipients. The method was introduced in previous literature [30]. Briefly, ENR, stearic acid, corn starch, and NaCl were sieved through a #80 screen. The drug was physically mixed with individual excipients in a 1:5 ratio. The samples were stored for one month at 40 °C/75% relative humidity and for two weeks at 60 °C in open scintillation vials. Finally, samples were analyzed by differential scanning calorimetry (DSC, DSC200PC, NETZSCH, Selb, Germany) and X-ray diffraction crystallography (XRD, Bruker AXS, Madison, WI, Germany). The data were analyzed by Origin (version 2017C, OriginLab Corporation, Northampton, MA, USA).

#### 2.3.2. Preparation of ENR-Loaded Stearic Acid SD

ENR-loaded stearic acid-SDs (ENR-SDs) were prepared by the hot-melt cold method, which was introduced in a previous study [31,32]. Briefly, a certain weight of ENR and stearic acid were added to a beaker and heated (100 °C) under stirring. After the ENR was completely dissolved into stearic acid, the hot melt solution was poured on the cold steel plate to let the hot solution quickly cool down, thus allowing the ENR to be dispersed into stearic acid. After cooling down for 30 min, the formed SDs were milled by a grinder (BJ-800A, Baijie, China). After sieving through a #80 screen, SD powders with different weight ratios were obtained. In further steps, solubility tests were performed to screen the optimal SD with satisfactory solubility and to clarify the relationship between solubility and the ratio of ENR and stearic acid.

#### 2.3.3. Characteristics of ENR-Loaded Stearic Acid-SD

The solubility of ENR-SDs with different ratios was detected by the maximum solubility method, which was reported by Kwon et al. [33]. Briefly, sufficient amounts of ENR-SD powder, prepared by different ratios of ENR and stearic acid, were added to brown bottles containing 100 mL of simulated intestinal fluid (SIF, 1000 mL SIF containing 6.8 g of KH_2_PO_3_ and 10 g of trypsin and then adjusting the pH to 6.5 with NaOH solution) buffer and shaken for five days (38.5 °C, 100 rpm). Five milliliters of the solution at 0.5, 3 and 5 d was sampled. The sampling suspensions were centrifuged (12,000 rpm, 10 min). Then, the supernatant was collected and filtered through a 0.25-μm filter. After dilution, the concentration of ENR was assayed by HPLC (Waters 2475, Milford, MA, USA). According to the solubilities of different ENR-SDs, the weight ratio of ENR and stearic acid that can produce a high-solubility ENR was selected, and the effects of the weight ratio on the solubility were discussed.

The thermal properties and optical patterns of the optimal ENR-SD powder and the excipients were analyzed by DSC and XRD, respectively. All measurements were repeated in triplicate using different samples from independent preparations.

#### 2.3.4. Preparation of ENR-SD Granules

ENR-SD granules containing ENR-SD powder were prepared by wet granulation (Figure 1). Briefly, the SD powder was completely mixed with corn starch and NaCl using a #80 screen. Then, starch paste (10%, *W*/*V*) was added to prepare a suitable soft material. Then, the soft material was granulated by a granulator (YK-160, ZhiYang Machinery CO., LTD, Changzhou, China), and wet granules were prepared. After the extra water from the wet granules was removed in a dry oven (GMP-O, ZhiYang Machinery CO., LTD. Changzhou, China) (40 °C, 24 h), ENR granules containing ENR-SD were obtained.

#### 2.3.5. In Vitro Release Test

The in vitro release of prepared ENR-SD granules and commercial ENR tablets was performed in simulated gastric fluid (SGF, 1000 mL containing 2.0 g of NaCl and 3.2 g of pepsin and then adjusting the pH to 2.0 with HCl) and SIF by a dissolution tester (RC8MD, TIANDA TIANFA Equipment Co., Ltd. Tianjin, China). Briefly, 1.0 g of ENR-SD granules (containing 50 mg of ENR) and two commercial ENR tablets (containing 50 mg of ENR) were placed in a dissolution cup containing 500 mL of SGF or SIF buffer. According to the body temperature (38~39.5 °C) and gastric emptying time (1.5~2 h) of pigs [34], the release system was sustained at 38.5 °C and 75 rpm, and the release time was 2 h. Five milliliters of fresh SGF or SIF was added after sampling to maintain a constant volume. Three parallel tests were performed in each group. The drug concentration was determined by HPLC. The cumulative release profiles were drawn in GraphPad Prism (version 7.0, GraphPad Software Inc., La Jolla, CA, USA). The cumulative release of ENR was determined using Equation (1) [35]:

Cumulative release of ENR (%) = Rt/L × 100
(1)
where L and Rt represent the initial amount of ENR loaded and the cumulative amount of ENR released at time t, respectively.

### 2.4. Palatability Test

Palatability is important for an oral product. To verify the palatability of prepared ENR granules for pigs, a feeding experiment was performed, which was introduced in detail in our previous work [18,36]. The eighteen pigs were divided into two groups (a control group and an ENR granule group). Three parallel tests were performed in each group. After free food intake was allowed for seven days, a sufficient amount of control feed or mixed with ENR granules (2.5 g/kg feed) was fed to the pigs at 9:00 a.m. Then, the remaining feed was collected on the next day at 9:00 a.m. The amount of intake was equal to the weight of added food after subtracting the weight of the rest. The amount of intake in the two groups was compared to verify the palatability of ENR-SD granules.

### 2.5. Development of the PBPK Model for ENR Granules in Pigs

The PBPK model in pigs of ENR granules was built with acslXtreme (version 3.0, The Aegis, Technologies Group, Inc. Huntsville, AK, USA). The physiological parameters of pigs and the chemical-specific parameters for ENR in pigs are from the literature [37,38,39,40,41] and are provided in Appendix A. The model included the oral administration module, gastrointestinal tract, plasma, liver, kidney, fat, muscle, lung, and the rest of the body (Figure 2). The small intestine was included since the local concentration in the small intestine was of interest in this study. The liver and kidney were modeled as individual compartments because these organs are responsible for metabolism and excretion. Plasma was included since it is an essential compartment linking all other compartments through the systemic circulation. Because ENR is also used to treat pneumonic infections (e.g., *Mycoplasma* pneumonia and pleuropneumonia), the lung was considered one compartment. All compartments were assumed to be blood flow-limited and well-stirred. Equations and complete modeling code describing the ENR absorption, distribution, metabolism, and elimination processes are provided in the Appendix A. The oral PBPK model of ENR in pigs was well introduced by Lin et al. [38], thereby, the modeling code and the main parameters used in our paper was cited from their paper. Meanwhile, the modeling code can be found in website of College of Veterinary Medicine, Kansas State University (http://iccm.k-state.edu/, accessed on 23 March 2021). Additionally, visually reasonable values for small intestinal volume and gastric emptying rate constant were obtained by an iterative manual adjustment approach to fit the experimental data for model calibration (Appendix A).

### 2.6. Validation of the PBPK Model

To verify the accuracy of the PBPK model prediction, the plasma PK experiment was performed with six healthy pigs. ENR granules (content: 5%) resuspended in 20 mL of CMCC-Na were administered to the pigs by intragastric administration with 2.5 mg/kg b.w. After drug administration, three-milliliter blood samples were collected at 0.25, 0.5, 0.75, 1, 1.5, 2, 4, 8, 12, 24, and 48 h, placed in centrifuge tubes with heparin sodium and centrifuged at 4000 rpm for 10 min to separate the plasma. The drug concentration in the plasma was detected by HPLC after pretreatment. PK parameters were calculated with WinNonlin software (version 6.4; Pharsight Corporation, Mountain View, CA, USA) using noncompartment analysis. Besides, the withdrawal time of the ENR-SD granules in pigs were performed in our later studies. Therefore, to reduce animal use, fifteen samples of intestinal contents at 1 h, 109 h, 112 h, 120 h, and 132 h were obtained from fifteen dissected pigs in our later tissue residue experiments, which were administered at a dose of 5 mg/kg b.w. twice per day for five days.

The plasma and intestinal content samples were detected by HPLC after pretreatment. Briefly, 0.8 mL of plasma was added into a 10-mL tube containing 1.6 mL of methanol. The plasma and methanol were thoroughly mixed under vortexing for 2 min to precipitate protein and were then centrifuged at 12,000 rpm for 20 min to obtain the supernatant. The supernatant was dried in nitrogen and reconstituted with 800 μL of methanol. The pretreatment of intestinal content samples was similar to the plasma samples. Intestinal content (0.1 mL) was added to a 10-mL tube containing 2.0 mL of methanol. After protein precipitation, centrifugation, and drying under nitrogen (same conditions as plasma), 1.9 mL of methanol was used for reconstitution. Then, the liquid of plasma samples and intestinal content samples was filtered through a 0.22-μm filter and detected by HPLC. A C_18_ column (SB-Aq, 250 × 4.6 mm, i.d., 5 μm, Agilent, Carpinteria, CA, USA) was used for HPLC, which was performed with an excitation wavelength of 280 nm and an emission wavelength of 450 nm. The mobile phase consisted of 0.5 M phosphoric acid mixed with triethylamine (phase A, pH = 2.4) and acetonitrile (phase B) at a ratio of 82: 18 (*V:V*). The ENR content was determined using a standard curve. The linear ranges of the standard curves for plasma and intestinal content ranged from 0.04 to 2.0 μg/mL (R^2^ = 0.9997) and from 0.2 to 10.0 μg/mL (R^2^ = 0.99), respectively. The limits of detection and quantitation for plasma were 0.02 and 0.04 μg/mL, respectively. The limits of detection and quantitation for intestinal content were both 0.04 μg/mL. The RSD of precision was less than 10%, and the recovery rates of three different added concentrations were 79.9–103.9%.

The drug concentrations of plasma and intestinal content were used as observed data to verify the predictive accuracy of the PBPK model. The predictive accuracy was defined by determination coefficients (R^2^ values) of linear regression analyses between model-predicted and experimental data using GraphPad Prism and two-fold error (FDA).

### 2.7. Model Application

To provide a guideline for the later clinical phase II, the potential doses against common intestine pathogen bacteria were predicted by the validated PBPK model. As mentioned above, for fluoroquinolones, the target parameter that better correlates to the therapeutic outcome is the AUC_24h_/MIC ratio. Reportedly, during therapy, for a given dose regimen of fluoroquinolones, the AUC_24h_/MIC ratio needs to be at least 100 for gram-negative bacterial infections [29,42]. Therefore, the target parameter was selected as AUC_24h_/MIC ≥ 100 in our study. After the model was validated by observed concentrations in the plasma and the intestinal contents, the drug concentration-time profiles of intestinal contents at different doses were extrapolated by changing the dose value in the model code. Then, the AUC_24h_ values in the intestinal contents (infection target site) at different doses were calculated based on the produced PK curves. To design a dose aimed at 90% of clinical bacterial strains, we collected the MIC_90_ values of ENR against swine *Salmonella*, *Camp. jejuni*, and *E. coli* from the literature. Subsequently, combining the AUC_24h_ values with the MIC_90_ values, the ratio of AUC_24h_/MIC_90_ was obtained. Whether the AUC_24h_/MIC_90_ was >100 was considered to provide a potential efficient therapeutic dose for later clinical phase II experiments. Meanwhile, the effect of diarrhea (commonly caused by intestinal infections) on PK profiles was analyzed by adjusting the fecal elimination rate constant in the model; thus, the efficient dose focused on the diseased pigs was discussed.

### 2.8. Parameter Sensitivity Analysis

To search for potential measures that could further enhance the ENR concentration in the intestinal contents, to reduce the consumption of ENR in the clinic, PSA based on the modeling parameters was conducted to determine the effects of each parameter on the C_max_ of ENR in intestinal contents by the embedded PSA module in acslXtreme software. The normalized sensitivity coefficient (NSC) was used to determine the sensitive parameters and was calculated using Equation (2):

NSC = Δr/r × p/Δp
(2)
where p is the initial parameter value, Δp is a 1% increase in the parameter value, r is the model output derived from the original parameter value, and Δr is the change in model output due to the 1% increase in the parameter value. Parameters with absolute values of /NSC/ ≥ 0.5 were considered highly sensitive, and those with absolute values of 0.5 > /NSC/ ≥ 0.2 were considered of medium sensitivity [43].

## 3. Results and Discussion

### 3.1. Compatibility between ENR and Excipients

The results of compatibility studies are shown in Figure 3. The DSC results showed that the physical mixture of ENR and stearic acid had a peak at 73.4 °C, suggesting that the melting point of stearic acid was approximately 73.4 °C, which agreed with the result of pure stearic acid DSC and the PubChem report (68~72 °C) (https://www.ncbi.nlm.nih.gov/pccompound, accessed on 29 June 2005). The mixture of ENR and stearic acid or ENR and NaCl had a peak at 225.0 °C, indicating that the melting point of ENR was approximately 225.0 °C, which also agreed with the report of PubChem. Because of the high melting point, no NaCl melting point (801 °C, PubChem) peak was observed. Since the samples were only heated up to 250°, the NaCl melting point could not be detected in the present method. In addition, no ENR peak in the stearic acid mixture was observed, which might be due to the formation of amorphous ENR in the presence of stearic acid at high DCS temperatures. The XRD results indicated that the most intense peaks in the XRD of the ENR powder were visible at 9.7 2-theta degrees, which agreed with previous studies [44]. Although the ENR peak was reduced in the binary physical mixture, the ENR peak did not disappear, suggesting that ENR was present as crystals during the compatibility tests (Figure 3B–D). The binary physical mixture of ENR with stearic acid and ENR with NaCl displays another intense peak, suggesting that the 2-theta degrees of stearic acid and NaCl were 21.6 and 29.4, respectively, which agreed with previous studies [45,46]. In addition, no extra peaks were observed in the DSC curves under 40 °C for 30 d or 60 °C for 14 d (Figure 3A), and the XRD peak shapes of ENR with stearic acid, ENR with starch, and ENR with NaCl at 40 °C for 30 d or 60 °C for 14 d were similar to those at 0 d (Figure 3B–D). These results demonstrated no specific interactions between the excipients and ENR under the compatibility experiments, suggesting that the compatibilities of ENR, stearic acid, starch, and NaCl were acceptable.

Due to the high content of stearic acid in the granule formulation, the toxicity of stearic acid should be discussed. Reportedly, the LD_50_ of acute oral toxicity to rats and the LD_50_ of acute dermal toxicity to rabbits are 4640 mg/kg and >5000 mg/kg for stearic acid, respectively (Drug Bank). In addition, Wang et al. [47] reported that stearic acid could protect cortical neurons against oxidative stress by boosting internal antioxidant enzymes. These results suggested that stearic acid is a low-toxicity compound. Meanwhile, due to the significantly higher melting point (73.4 °C) of stearic acid than the common ambient temperature, good storage stability of ENR-SD granules could be expected. Additionally, considering its high solubility to ENR and its low price, [18] stearic acid was selected as the SD material in our study.

### 3.2. Solubility of ENR in SDs

The saturated solubilities (38.5 °C, pH = 6.8) of ENR-SDs with different weight ratios of ENR and stearic acid are shown in Figure 4. The saturated solubility of ENR in the neutral condition was 506 ± 6.94 µg/mL, which agrees with a previous study (490 µg/mL) [17]. Therefore, the saturated solubility data in this paper were reliable. When the weight ratio was 1:4, the largest saturated solubility of ENR was achieved, measured at 2206 ± 17.45 µg/mL. When the weight ratio was increased to 1:5, the saturated solubility of ENR was 1190 ± 7.71 µg/mL. Compared with native ENR, the saturated solubility of ENR in the neutral condition was improved 4.36- and 2.35-fold, respectively. Although the saturated solubility improvement by a weight ratio of 1:4 was more than that of 1:5, the former needed to be heated to approximately 130 °C to completely dissolve the ENR, and the temperature of the latter was approximately 100 °C. The higher temperature requires more power, and it was difficult to sustain 130 °C in our pilot workshop. Therefore, a weight ratio of 1:5 was used in the following studies to collect formulation strategy data.

Increasing the weight ratio to 1:8 and 1:10 did not improve the saturated solubility of ENR, which suggested that the enhancement level of solubility is closely related to the weight ratio of the drug and the SD materials. Kwon et al. [33] indicated that when the ratio of atorvastatin calcium (ATO), hydroxypropyl methylcellulose (HPMC) and sodium lauryl sulfate was 1:1:0.1, the largest saturated aqueous solubility of ATO could be obtained. Increasing the weight ratio of ATO and HPMC decreased the saturated solubility. In addition to the spray-dried method, the solvent and hot-melt methods have been commonly adopted to prepare SDs [48,49,50]. Compared with the spray-dried and solvent methods, the hot-melt method requires fewer operation skills and produces less pollution in the environment. Therefore, to facilitate pilot production, the hot-melt method was adopted in our study. Additionally, the saturated solubility of ENR-SDs was not changed within 5 days (*p* < 0.05), suggesting that no ENR was recrystallized in its supersaturation solution and that its stability against crystallization of ENR-SDs was strong [51]. These results indicated that the solubility of ENR was improved by ENR-SD and that a higher solubility of ENR-SD in the intestinal contents could be achieved. Additionally, due to the strong ability of ENR-SD against recrystallize, the wet granulation method was adopted in the present study.

### 3.3. Drug Crystals Form in the ENR-SD

The existing state of the drug in SD was closely related to its in vivo performance. The ENR crystals in the optimal ENR-SD (weight ratio of 1:5) were analyzed by DSC and XRD, and the results are shown in Figure 5. The DSC results of stearic acid and ENR revealed the presence of single peaks of stearic acid and ENR, suggesting that the melting points of stearic acid and enrofloxacin were approximately 73.4 °C and 225 °C, respectively. For ENR-SD, there was only the endothermic peak of stearic acid in its DSC curve (Figure 5A). Meanwhile, the XRD results indicated that the intense peaks of ENR and stearic acid were observed at 9.7 and 22.2 2-theta, respectively. However, only the stearic acid peak was shown in the ENR-SD curve (Figure 5B). Therefore, both the DSC and XRD results of ENR-SD proved that ENR presented as an amorphous form in ENR-SD [36].

### 3.4. In Vitro Release Performance of ENR-SD Granules

The in vitro release profiles of the commercial tablets (approved for pets) and the prepared ENR-SD granules in SGF and SIF are shown in Figure 6. In the SGF, 97.72% of the commercial tablet was released within 45 min, while 72.74% of the prepared ENR-SD granules were released. Due to the encapsulation of stearic acid, the corrosion of stearic acid delayed the release of ENR in the ENR-SD granules, thus leading to the slightly slower release rate of ENR-SD granules. In the SIF, compared with the 95.0% release of ENR within 90 min, 85.05% of the commercial tablets released, suggesting that the prepared ENR-SD granule would release slightly more quickly than the commercial tablet in the SIF. As mentioned above, the gastric emptying time of pigs was 1.5~2 h, and the small intestine transit time of pigs was 3–4 h [34]; therefore, ENR-SD granules could be completely released in the small intestine to ensure a high C_max_ and a short T_max_ in the intestinal contents. Meanwhile, due to the larger saturated solubility of ENR under acidic conditions than under neutral conditions, both the tablet and the ENR-SD granule showed faster release rates in SGF than in SIF.

Overall, the prepared ENR granules displayed similar in vitro release performance to successful commercial tablets, suggesting their potential in vivo PK. In addition, the low release rate of ENR-SD granules within 15 min indicated the relatively complete encapsulation of ENR by stearic acid, which suggested that less free ENR would contact the taste buds of pigs when the ENR-SD granules were fed by pigs. Because ENR is a strong bitterness drug for pigs, for its taste enhancement, its taste during chewing by pigs also needs to be considered. Although some soluble materials (e.g., polyethylene glycol, polyvinyl alcohol, HPMC and poloxamer) could be used to prepare SDs, the drug SDs prepared by soluble materials commonly showed very quick dissolution [52,53], thus enhancing the contact between drugs and buds of the pigs. Therefore, to ensure the taste-masking effect, a water-insoluble material rather than a soluble material was chosen in our study.

### 3.5. Palatability of ENR-SD Granules

The daily feed intake rates of pigs before and during the palatability experiment are shown in Table 1. The daily feed intake rates of pigs in the two groups were normal and close (1.25–1.37 kg/day) to each other before the palatability experiment. This proved that the selected pigs were healthy and physiologically consistent. The average daily feed intake rates of pigs in the control group and ENR granule group were 1.38 ± 0.03 kg/day and 1.36 ± 0.03 kg/day, respectively, suggesting that the palatability of prepared ENR-SD granules was acceptable and that mixing the prepared ENR-SD granules into feed would not influence the drug intake. There is no doubt that the prepared ENR-SD granules with good palatability will be advantageous for the clinical application of ENR. Meanwhile, the poor palatability of ENR is well known and was proven in our previous work. The pig daily feed intake in mixing the native ENR group was 0.15 ± 0.03 kg/day [18]. Comparing the feed intake in the ENR-SD granule group (1.36 ± 0.03 kg/day) and that of the mixing native ENR, ENR-SD effectively improved the palatability of ENR. As mentioned above, the enhancement of ENR palatability might be due to the reduced release of ENR in the mouth and the masking effect of NaCl.

### 3.6. Pharmacokinetics and the Validation of the PBPK Model

After the palatability of prepared ENR-SD granules was verified, the in vivo PK assessment of the ENR-SD granules was performed. As shown in Table 2, the AUC_0-last_, C_max_, and T_max_ of ENR-SD granules in plasma were 7.96 ± 1.22 µg h/mL, 0.64 ± 0.21 µg/mL, and 1.42 ± 0.34 h, respectively. Our previous study demonstrated that the AUC_0-last_, C_max_, and T_max_ of commercial ENR soluble powder at the same dose in pig plasma were 4.26 ± 0.85 μg h/mL, 0.60 ± 0.12 μg/mL, and 1.12 ± 0.44 h, respectively [18]. Compared with the commercial ENR soluble powder (present as solution in the intestine), their C_max_ and T_max_ values were close, suggesting similar absorption performance between prepared ENR-SD granules and the commercial soluble powder. Therefore, we argued that ENR within ENR-SD granules was also present as a solution when it arrived in the intestine. However, this indicated that to continue to enhance the saturated solubility of ENR, a higher drug concentration in the intestinal contents would not be obtained. In other words, the aqueous solubility of ENR would not limit its C_max_ and T_max_ in the intestinal contents when ENR was prepared at a 1:5 ratio; thus, the weight ratio of ENR and stearic acid was determined to be 1:5, and the heating temperature was determined to be 100 °C. It seems that the oral bioavailability of prepared ENR-SD granules was slightly higher than that of commercial soluble powder. However, because of the different limits of quantitation of the two HPLC detection methods, conclusions could not be drawn. In other words, ENR-SD improved the solubility and palatability of ENR but did not obviously decrease the local intestinal drug concentration compared with commercial soluble powder.

Besides, our previous studies proved that the AUC_0-last_, C_max_ and T_max_ of enteric ENR granules containing ENR nanoparticles in pig plasma at a dose of 2.5 mg/b.w. kg were 11.24 ± 3.33 μg h/mL, 0.52 ± 0.02 μg/mL and 3.33 ± 1.03 h, respectively [18]. Meanwhile, Lei et al. [54] provided an ENR enteric-coated granule. Plasma PK results showed that the T_max_ of the ENR enteric-coated granules in the plasma of pigs was 3.99 ± 0.10 h at a dose of 10 mg/kg b.w. The longer T_max_ in the plasma of the two enteric granules than that of the ENR-SD granules suggested a slower release rate of enteric coating granules in the intestine. Obviously, although enteric coating technology was helpful for improving the palatability of drugs, the concentration-dependent antibiotic products prepared by enteric coating were not suitable for treating intestinal infections. A longer exposure time of sub-antibacterial concentrations means a higher mutation probability for bacteria. Obviously, for intestinal infections, a technology that could improve both the palatability and solubility of ENR is needed. Therefore, drug amorphous SD technology was adopted in our study. The in vitro release, palatability, and in vivo PK results demonstrated that ENR-SD granules with good palatability for pigs and high drug concentrations in intestinal contents were successfully prepared. Additionally, the preparation processes of ENR-SD granules were simple and convenient for large-scale production.

After the plasma drug concentration was detected, the predictive plasma concentration was verified by observed plasma data. As shown in Figure 7, the predicted plasma drug concentration data were well fitted by the observed data (Figure 7A). Linear regression analysis indicated that the (R^2^) value was 0.9285 (Figure 7B), suggesting adequate goodness-of-fit of the model. This proved that our model could be used to predict the concentrations of ENR in the plasma and other tissues (e.g., intestinal contents) of pigs at different doses with acceptable accuracy. Therefore, for the further development of ENR-SD granules, the potential reference dose regimen for later clinical phase II was predicted by combining the predicted intestinal PK parameter and MIC values. This is meaningful for reducing animal use in dose design and tissue residue experiments.

### 3.7. Dose Prediction of ENR-SD Granules

After the constructed model was verified, the PBPK model was applied to predict the ENR concentration in intestinal contents versus time at a dose of 5 mg/kg b.w. twice a day for five days. The predicted value was well verified by five-measured data (R^2^ = 0.9791), suggesting the strong extrapolation power of the PBPK model (Figure 8A,B). To provide the reference dose for later clinical phase II experiments, the potential therapeutic efficacy at different doses was predicted by combining the PK and MIC values. The MIC_90_ values of ENR against swine *Camp. jejuni*, *Salmonella*, *E. coli*, and *Lawsonia intracellularis* (*L. intracellularis*) that were collected from the literature are provided in Table 3 [10,11,55,56,57]. According to the predicted PK curve, the AUC_24h_, C_max_, and T_max_ of ENR in intestinal contents were 1065.23 μg h/mL, 201.07 μg/mL, and 1.0 h, respectively, at a dose of 5 mg/kg b.w. once daily. The AUC_24h_/MIC_90_ ratios against *Salmonella*, *E. coli*, and *L. intracellularis* were 133, 4260 and 133, respectively. When increasing the dose to 10 mg/kg b.w. once daily, the AUC_24h_ in intestinal content was 2130 μg h/mL, and the AUC_24h_/MIC_90_ against *Camp. jejuni* was 133. As mentioned above, when the AUC_24h_/MIC_90_ ratio of fluoroquinolones was >100, the bacteria could be inhibited [29,42]. These results indicated that when the prepared ENR-SD granules were orally administered at a dose of 5 mg/kg b.w. once daily, swine intestinal infections caused by *Salmonella*, *E. coli*, and *L. intracellularis* could be well treated. In addition, the *Camp. jejuni* infections could also be controlled at a dose of 10 mg/kg b.w. once daily. Furthermore, a hollow fiber infection model indicated that when the AUC_24h_/MIC_90_ of ENR was 289.64, the resistant subpopulations of swine *Salmonella* could be inhibited [58], serving as a reminder that when the dose of ENR-SD granules was approximately 10 mg/kg b.w. once daily (AUC_24h_/MIC_90_ = 266), not only could the infections be treated but also the occurrence of the resistant subpopulation could be inhibited. Importantly, although the AUC_24h_ of 5 mg/kg b.w. twice per daily was close to the 10 mg/kg b.w. once daily (Table 3), an obvious drug peak and valley were observed in the former (Figure 8C). Considering the resistance mutation selection window theory, the 10 mg/kg b.w. once daily dosage was the better dose regimen. This met the principle of concentrating the dose for concentration-dependent antibiotics [42].

Notably, to obtain intestinal content samples, abdominal surgery needs to be performed on pigs. This not only requires more animal use but also causes animal welfare issues. One of the advantages of the PBPK model is that we can make use of it to predict the drug concentration in tissues that are hard to sample, thus reducing animal use during drug development [59]. In addition, the drug concentration in target tissues was verified by five data points in the present study. In contrast, Lin et al. [38] verified the predicted concentration of sulfamethazine in cattle muscle by only one data point. Yet in this study, the timepoints of the absorption and elimination phases of PK curves were both included in our study. Therefore, we believe that the predicted PK curves in intestinal contents are reliable. The satisfactory fitting results might be due to the selected modeling parameters were close to the experimental values (Appendix A). Obviously, the combination of AUC_24h_ and MIC_90_ provided a reference dose for later clinical phase II experiments. To the best of our knowledge, this is the first study to construct a synchronous PBPK-PD model in the field of veterinary drugs.

For further efficacy verification of the above dose regimen, we will combine the PBPK model with a hollow fiber model to build a PBPK-PD model in our later studies, thus combining the predictive PK curves of intestinal contents with the growth equation of bacteria to verify the efficacy of the above dose regimens. Kuepfer et al. [60] proved that the effective dose regimen of ciprofloxacin against pulmonary interstitial *E. coli* infection was 1000 mg once daily for two days by a PBPK-PD model of ciprofloxacin against *E. coli*. In addition, the inhibitory effect on resistant subpopulations could also be verified by a hollow fiber model [61]. Therefore, whether the above predictive dose regimens could reduce the emergence of the resistant subpopulation could also be further analyzed by isolating the resistant subpopulation in the hollow fiber model. In addition to the advantage of reducing animal use, a dose regimen against intracellular pathogens could also be designed using the hollow fiber model [62]. This suggested the feasibility of designing a dose regimen of ENR-SD granules against intracellular *Salmonella* infection.

Importantly, the high MIC_90_ values of *Camp. jejuni* and *Salmonella* served as reminders that prevention resistance methods needed to be adopted to delay the resistance development of *Camp. jejuni* and *Salmonella*. It was reported that because of the inhibitory activity on the functional membrane microdomain assembly of statins, after statin treatment, the penicillin-binding protein PBP2a, which confers *β*-lactam resistance to MRSA, was inhibited; thus, MRSA became susceptible to conventional antibiotic agents [63]. Additionally, due to their pro-inflammatory response downregulation and antibacterial activity enhancement actions, statins were recommended as an adjunctive treatment in *Salmonella* infection or inflammatory bowel disease by Huang et al. [64]. Whether ENR-resistant *Salmonella* or *Camp. jejuni* could become susceptible after statin treatment, and whether the combination of a statin with ENR could improve the antibacterial efficiency against *Salmonella* or *Camp. jejuni* could also be directly verified by detecting the bacterial count and resistance genes or protein expression level in the hollow fiber model. Reportedly, the hollow fiber model is also helpful in drug compound design [65,66]. Therefore, the in vitro efficient concentration of statins could be reversed to a related in vivo dose by the PBPK model; thus, an ideal ENR-statin compound focused on intestinal infections and intestinal resistant pathogens could be well designed.

### 3.8. Sensitivity Parameters and Development Strategies of ENR-SD Granules

When the PBPK model was verified, PSA was performed to determine the effects of modeling parameters on the ENR peak concentration in intestinal contents to search for potential formulation strategies for achieving a higher peak concentration. After administration, the ENR was directly exposed to the small intestine. Therefore, only the BW, Ka, Kst, and Kfeces related to intestinal absorption and transmission were analyzed in the present study. As shown in Figure 9, the ENR concentration in intestinal contents was moderately sensitive to feces (/NSC/ = 0.2), suggesting that diarrhea will reduce the ENR peak concentration in the intestine. Therefore, for pigs with diarrhea, a higher dose was needed to achieve satisfactory efficacy. The ENR peak concentration in intestinal contents was highly sensitive to Ka (/NSC/ = 4.9). Ka represents the absorption rate of the intestine; the PSA results indicated that a lower Ka value will produce a higher ENR peak concentration in the intestinal contents, which remined us that we could obtain a higher peak ENR concentration in the local intestinal contents by reducing the Ka.

P-glycoprotein (P-gp) in intestinal epithelial cells limits the absorption of P-gp substrate drugs from the intestine, reminding us that we could develop an ENR-SD granule product containing P-gp inducer agents to further enhance the ENR peak concentration in intestinal contents. Then, higher efficacy could be obtained with less ENR use with this intestinal infection-specific ENR product. Reportedly, quercetin (a P-gp inducer) decreased the AUC and C_max_ of plasma ENR in chickens by increasing P-gp expression [67]. P-gp is an absorptive agent in the skin, and the inhibition of P-gp represents a strategy to promote cutaneous localization. Giacone et al. [68] proved that a nanoemulsion containing elacridar (P-gp inhibitor) showed a higher epidermal local concentration and lower systemic exposure. Similarly, the inducers in ENR-SD granules can improve the expression of P-gp in intestinal cells, thus limiting the absorption of ENR from the intestine and enhancing the intestinal local ENR concentration. This will be beneficial for reducing the usage of ENR and delaying the development of resistance to ENR in intestinal bacteria. The constructed PBPK model provided further formulation optimization strategies for the development of ENR-SD granules.

## 4. Conclusions

To prevent and control swine intestinal infections caused by *Camp. jejuni*, *Salmonella* spp., and *E. coli* and to reduce the threats caused by zoonotic intestinal pathogens to humans from the source, ENR-SD granules with good palatability for pigs and fast dissolution in the intestinal contents were prepared using amorphous SD technology. The DSC and XRD results showed that no specific interactions existed between the excipients and ENR during the compatibility tests, and ENR presented as an amorphous form in ENR-SD. The saturated aqueous solubility of ENR was enhanced by 2.35-fold compared with that of native ENR, and the enhancement level was related to the ratio of ENR and stearic acid. Based on the similar C_max_ and T_max_ values of ENR-SD granules and the commercial ENR soluble powder (present as solution in the intestine) in the plasma, suggesting it continued to enhance the solubility of ENR, the higher drug concentration in intestinal contents could not be obtained. After the PBPK model was verified by observed plasma and intestinal content data, the PK parameters in the intestinal contents (infection target site) at different doses were predicted. Combining the intestinal AUC_24h_ values and the MIC_90_ values, when the ENR-SD granules were administered at a dose of 10 mg/kg b.w. once daily, intestinal infections in pigs caused by the *Camp. jejuni*, *Salmonella* spp. and *E. coli* might be well treated. The PSA results proved that the concentration of ENR in the intestinal content was negatively correlated with Ka, which suggested that we could further enhance the intestinal AUC_24h_ by adding P-gp inducer agents. Our studies proved that the constructed PBPK model provided not only formulation strategies for the further development of ENR-SD granules but also a reference dose regimen for clinical phase II trials. This is the first study to apply the PBPK model method to the field of veterinary drug development.

## Figures and Tables

**Figure 1 pharmaceutics-13-00602-f001:**
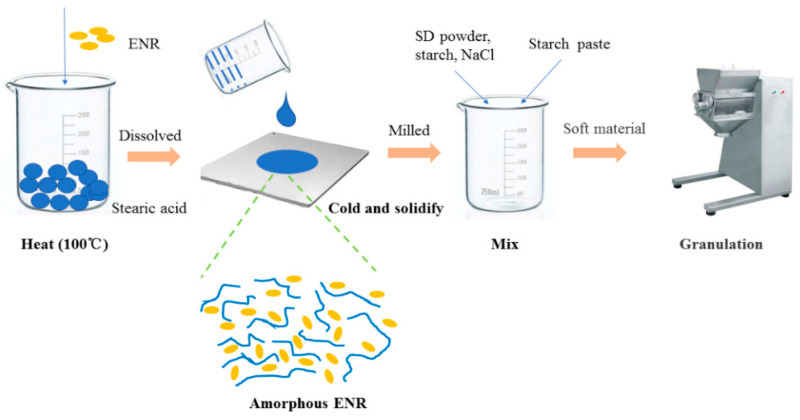
The productive process of the ENR granules.

**Figure 2 pharmaceutics-13-00602-f002:**
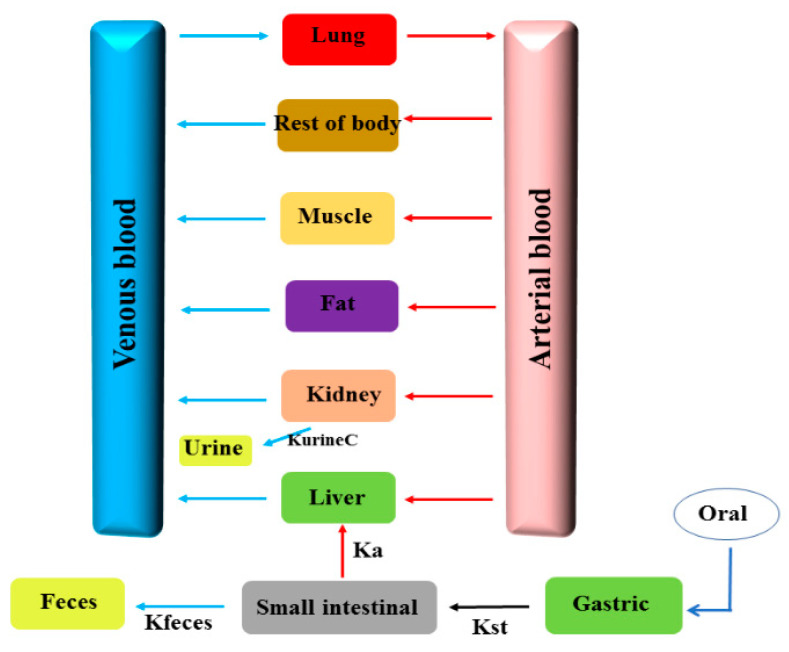
The schematic of the PBPK model for ENR in pig.

**Figure 3 pharmaceutics-13-00602-f003:**
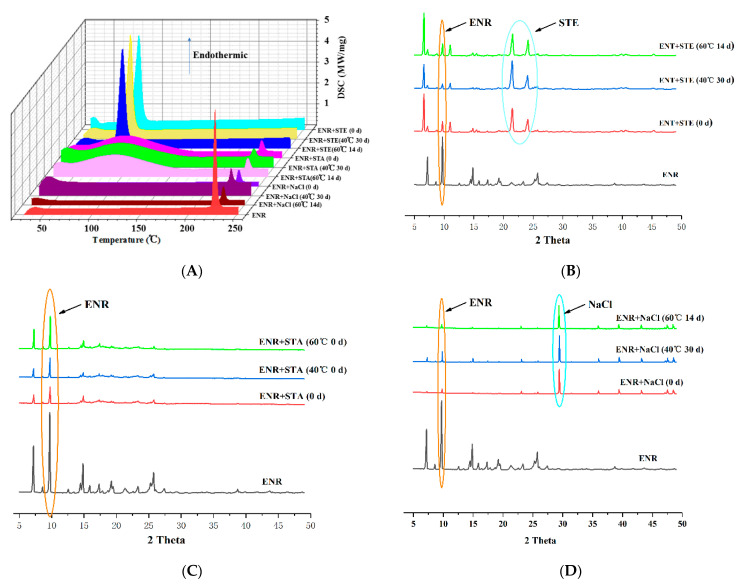
The compatibility of DSC (**A**) and XRD (**B**–**D**). No specific interactions exist between the excipients and ENR during the compatibility tests. Note: ENR, enrofloxacin; STE, stearic acid; STA, starch.

**Figure 4 pharmaceutics-13-00602-f004:**
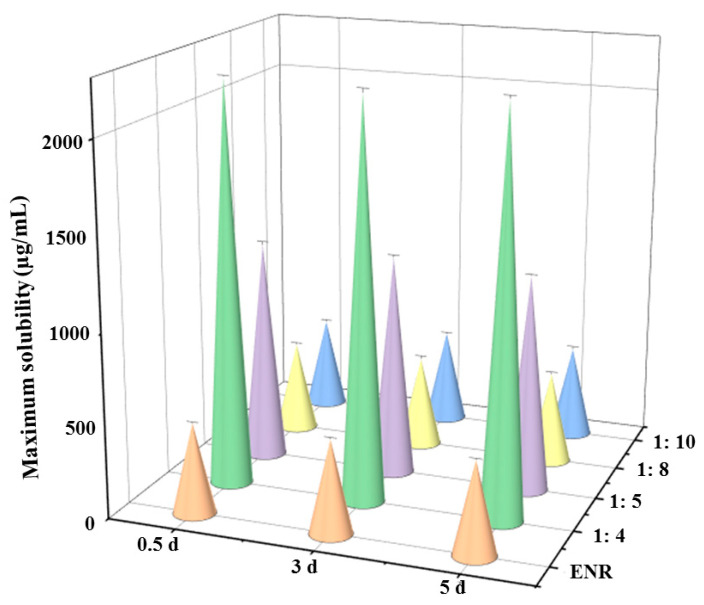
The solubility of ENR and different ENR-SDs (38.5 °C, pH = 6.8). The saturated solubility of ENR was enhanced 2.35-fold by ENR-SD (weight ratio 1:5). Note: ENR, native enrofloxacin, 1:4, 1:5, 1:8, 1:10, the weight ratio of ENR and stearic acid.

**Figure 5 pharmaceutics-13-00602-f005:**
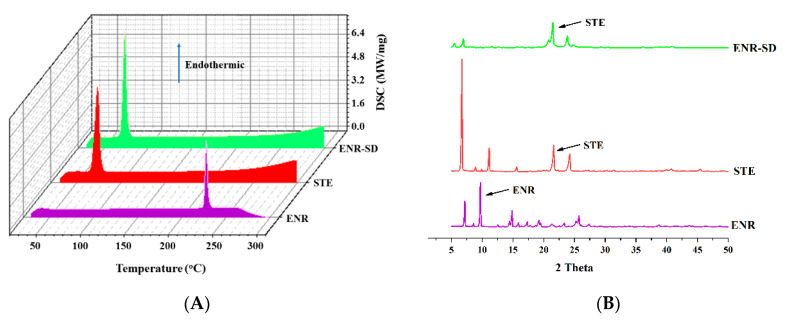
The DSC (**A**) and the XRD (**B**) analysis of ENR-SD. The ENR presented as an amorphous form in the ENR-SD. Note: ENR, enrofloxacin; STE, stearic acid; STA, starch.

**Figure 6 pharmaceutics-13-00602-f006:**
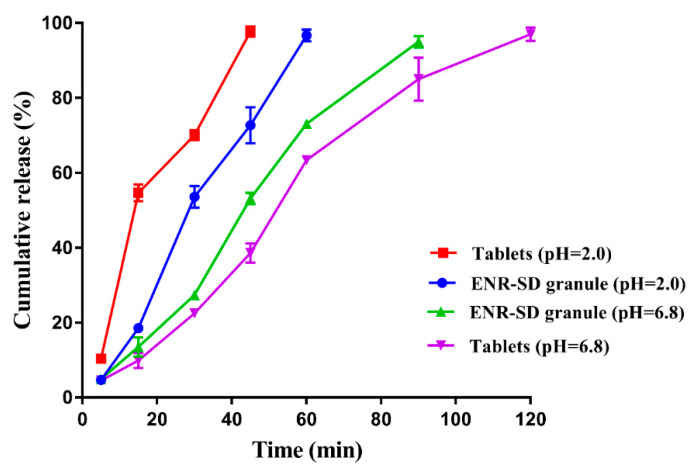
The in vitro release profiles of the commercial tablets and the ENR-SD granules in the SGF and the SIF. The ENR-SD granules showed a faster dissolution rate than that of commercial tablets in the SIF. Note: ENR-SD, enrofloxacin-loaded stearic acid solid dispersion.

**Figure 7 pharmaceutics-13-00602-f007:**
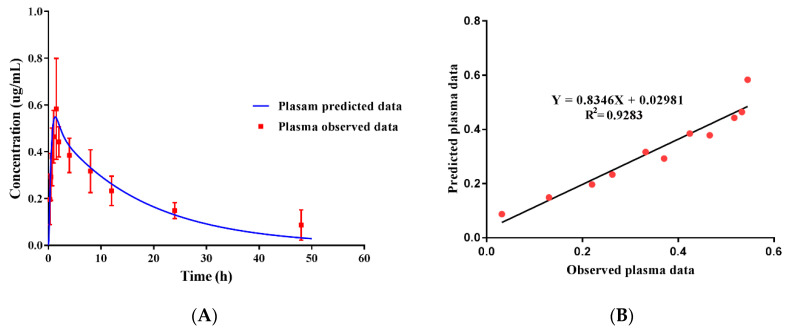
The fitting of observed plasma data with predicted data (**A**) and regression analysis of the PBPK model (**B**). The predicted plasma concentration was well fitted by the observed concentration (*r* = 0.9635).

**Figure 8 pharmaceutics-13-00602-f008:**
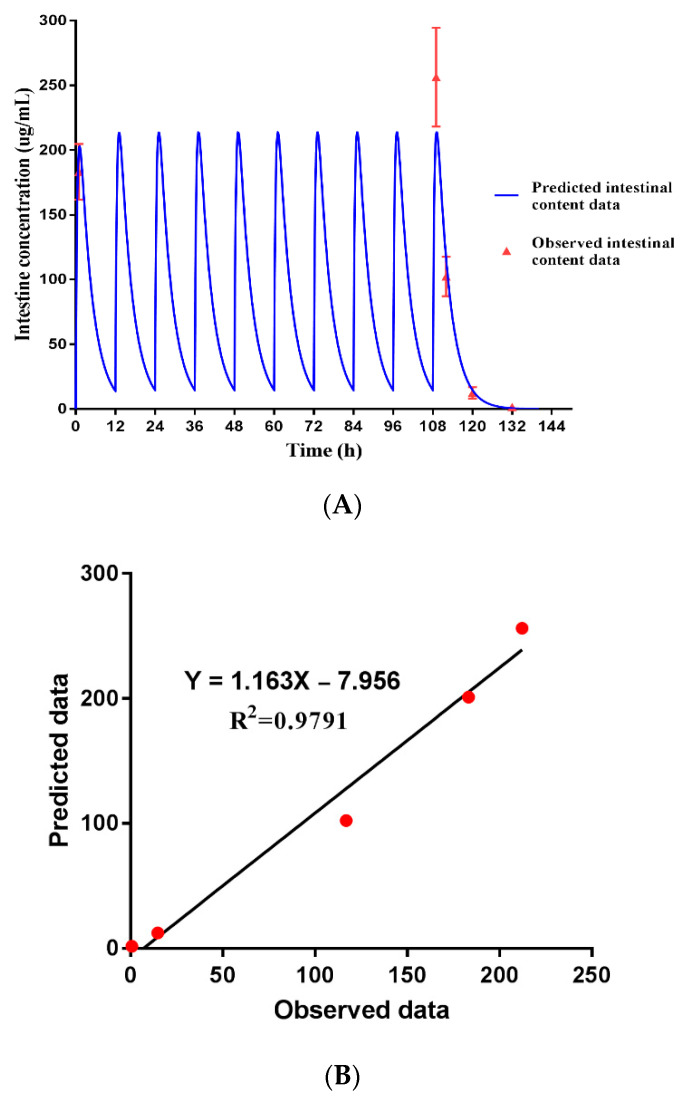
The fitting of observed ENR concentration in intestinal contents data with predicted data of the PBPK model (5 mg/kg b.w. twice a day for five days) (**A**), the liner regression analysis of the PBPK mode (**B**), and the predictive PK curves at different doses (**C**). The predicted ENR concentrations in intestinal content were well verified by five-observed data (*r* = 0.9894).

**Figure 9 pharmaceutics-13-00602-f009:**
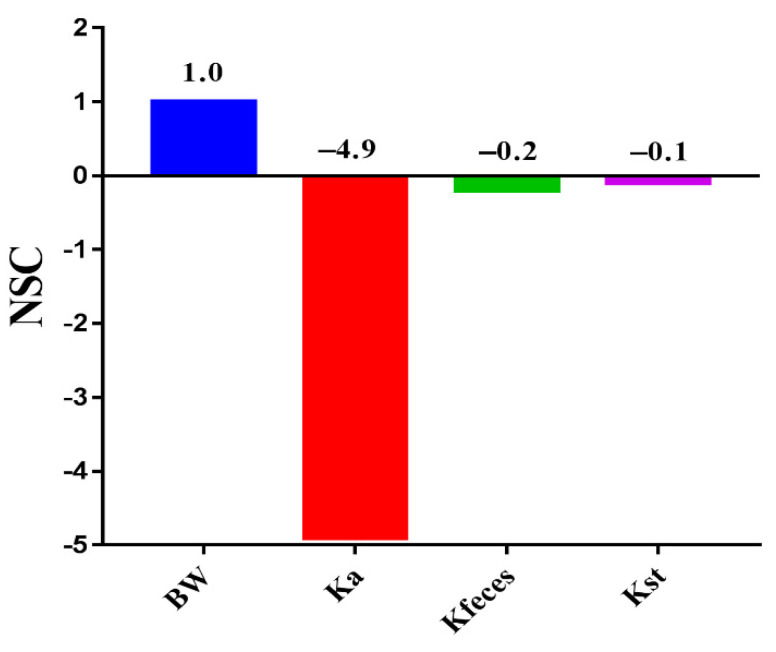
The sensitive parameters identified by the local sensitivity analysis. The ENR concentration in the intestine was highly sensitive to Ka. Note: BW, body weight; Ka, intestine absorption rate; Kfeces, feces elimination rate; Kst, gastric emptying rate.

**Table 1 pharmaceutics-13-00602-t001:** The daily feed intake of pigs in various groups (Mean ± SD, *n* = 3).

Groups	Before Experiment (kg/Group)				During Experiment (kg/Group)					
	1d	2d	3d	$\bar{x}$ ± SD	1d	2d	3d	4d	5d	$\bar{x}$ ± SD
Control	1.25	1.32	1.37	1.31 ± 0.05	1.30	1.28	1.35	1.38	1.42	1.38 ± 0.03
ENR granules	1.30	1.35	1.33	1.33 ± 0.02	1.25	1.30	1.32	1.37	1.40	1.36 ± 0.03

**Table 2 pharmaceutics-13-00602-t002:** Pharmacokinetic parameters for ENR-SD granules after oral administration in pigs (2.5 mg/kg b.w. once daily, *n* = 6, Mean ± SD).

Parameters	Unit	Value
Vd	mL/Kg	30,568 ± 15,686
MRT	h	15.37 ± 4.55
AUC_0-last_	µg h/mL	7.96 ± 1.22
T_1/2β_	h	12.58 ± 5.89
T_max_	h	1.42 ± 0.34
C_max_	µg/mL	0.64 ± 0.21

Note: Vd, apparent volume of distribution; MRT, mean residence time AUC, the area under drug concentration curve, T_1/2β_, half-life; T_max_, time to peak concentration; C_max_, peak concentration.

**Table 3 pharmaceutics-13-00602-t003:** The potential dose regimen of ENR-SD granules against common intestine bacterial in pigs.

Pathogens	MIC_90_ (µg/mL)	Dose Regimen	AUC_24h_ (µg h/mL)	AUCI
*Salmonella* (*n* = 291) ^a,b^	8.0	2.5 mg/kg, twice/day	957	120
*Salmonella* (*n* = 291) ^a,b^	8.0	5.0 mg/kg, once/day	1065	133
*E. coli* (*n* = 918) ^c^	0.25	5.0 mg/kg, once/day	1065	4260
*L. intracellularis* (*n* = 1) ^d^	8.0	5.0 mg/kg, once/day	1065	133
*Camp. Jejuni* (*n* = 114) ^e^	16.0	7.5 mg/kg, once/day	1542	96
*Camp. Jejuni* (*n* = 114) ^e^	16.0	5.0 mg/kg, twice/day	1884	118
*Camp. Jejuni* (*n* = 114) ^e^	16.0	10.0 mg/kg, once/day	2130	133

Note: ^a^ Hao et al., 2013; ^b^ Cao et al., 2017; ^c^ Wang et al., 2016; ^d^
*Lawsonia intracellularis*, Wattanaphansak et al., 2019, intracellular MIC, only one isolated; ^e^ Shin et al., 2007; AUCI, the ratio of AUC_24h_/MIC_90_; All bacterial were isolated from swine.

## Data Availability

Not applicable.

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
