# Peer review of "Application of a Physiologically Based Pharmacokinetic Model to Develop a Veterinary Amorphous Enrofloxacin Solid Dispersion"

_pharmaceutics, 2021, doi:10.3390/pharmaceutics13050602_

Round 1

Reviewer 1 Report

Dear authors, it is an interesting article.

My comments:

The introduction is too long; it should be shorter.

There are some abbreviations not explained (e.g., CMCC ...)

It is not clear, in the rows 680: how is it possible to obtain samples from "fifteen dissected pigs", if the PK experiment was "performed with six healthy pigs" (as mentioned in row 672).

Author Response

Reviewer#1: Dear authors, it is an interesting article.

Comment: 1. The introduction is too long; it should be shorter.

Response: Thank you for your comments. The introduction has been shortened in the revised manuscript (line 48-53, line 86-87).

Comment: 2. There are some abbreviations not explained (e.g., CMCC ...)

Response: CMCC-Na is the abbreviation of sodium carboxymethyl cellulose. I am sorry for the mistakes. Meanwhile, more abbreviations were defined in the revised manuscript (line 80 and line 119-125).

Comment: 3. It is not clear, in the rows 680: how is it possible to obtain samples from "fifteen dissected pigs", if the PK experiment was "performed with six healthy pigs" (as mentioned in row 672).

Response: Thank you for your comment. Actually, as what proposed in line 240, the fifteen samples of intestinal contents were obtained from fifteen dissected pigs in our later tissue residue experiments rather than the six healthy pigs that were used for plasma PK experiment. As we known, during the tissue residue experiments, the pigs need to be slaughtered to collect the edible tissues of pigs. Therefore, to reduce the animal usage, the intestinal contents were obtained from the pigs that were killed in the tissue residue experiments. I am sorry for the mistake that caused by the unclear descriptions. And the related descriptions were improved in the revised manuscript (line 238-240). The comments from you are significant helpful to improve the manuscript. Thank you for your guidelines sincerely.

Reviewer 2 Report

The authors have responded to my comments and the manuscript appears to be suitable for publication.

Author Response

Reviewer#2: The authors have responded to my comments and the manuscript appears to be suitable for publication.

Response: The comments from you are significant helpful to improve the manuscript. Thank you for your guidelines sincerely.

Round 2

Reviewer 1 Report

Dear authors,

The manuscript improved. Thank you for answering to all the comments.